# Sequence-level Intrinsic Exploration Model for Partially Observable Domains

## Abstract

Training reinforcement learning policies in partially observable domains with sparse reward signal is an important and open problem for the research community. In this paper, we introduce a new sequence-level intrinsic novelty model to tackle the challenge of training reinforcement learning policies in sparse rewarded partially observable domains. First, we propose a new reasoning paradigm to infer the novelty for the partially observable states, which is built upon forward dynamics prediction. Different from conventional approaches that perform self-prediction or one-step forward prediction, our proposed approach engages open-loop multi-step prediction, which enables the difficulty of novelty prediction to flexibly scale and thus results in high-quality novelty scores. Second, we propose a novel dual-LSTM architecture to facilitate the sequence-level reasoning over the partially observable state space. Our proposed architecture efficiently synthesizes information from an observation sequence and an action sequence to derive meaningful latent representations for inferring the novelty for states. To evaluate the efficiency of our proposed approach, we conduct extensive experiments on several challenging 3D navigation tasks from *ViZDoom* and *DeepMind Lab*. We also present results on two hard-exploration domains from Atari 2600 series in Appendix to demonstrate our proposed approach could generalize beyond partially observable navigation tasks. Overall, the experiment results reveal that our proposed intrinsic novelty model could outperform several state-of-the-art curiosity baselines with considerable significance in the testified domains.

## 1 Introduction

Under the reinforcement learning formalism, the learning behavior of an agent is driven by the reward that the agent collects from the environment (Sutton and Barto, 1998). However, many real-world problems have sparse rewards and most existing algorithms struggle with such sparsity. One inherent reason that leads to the inferior performance of the conventional approaches in sparse reward domains is that initially, the agent trained with those approaches could hardly stumble into a reward/goal state by chance due to their simple exploration strategies (Pathak *et al.*, 2017).

To tackle the sparse reward problems, it is crucial to incentivize the agent's exploration behavior. One prominent line of solutions for encouraging agent's exploration is via *reward shaping* (Singh, 1992; Dorigo and Colombetti, 1994), where the agent develops internal reward models to assign additional reward signals apart from the environment reward to encourage exploration. To model the internal reward signal, often, the agent's curiosity-driven behaviors are formalized as intrinsic novelty models (Schmidhuber, 1991; Singh *et al.*, 2004; Oudeyer *et al.*, 2007), which characterize agent's experience to compute the novelty scores.

Our work belongs to the broad category of methods that solve the sparse reward problems with novelty models and reward shaping. Specifically, we consider the line of sparse reward problems that employ partially observable inputs, with the inputs scaling to high-dimensional state spaces, such as images. Such problems cover a range of important applications among AI research, e.g., navigation, robotics control and video game playing. Even though the recently emerged intrinsic novelty models have demonstrate considerable efficiency in solving sparse reward problems with partial observability, we still face the following two major challenges. First, inferring the novelty for the true state given only the partial observations still remains an open problem. Most of today's

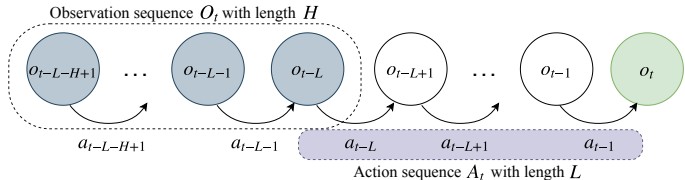

Figure 1: A high-level depict for the proposed reasoning paradigm of inferring novelty from multi-step forward dynamics prediction. A {L+H}-step transition graph is shown. The prediction of $o_t$ depends on a sequence of observations with length H followed by a sequence of actions with length L. Generally, the longer L, the more difficult to predict $o_t$.

state-of-the-art novelty models (e.g., (Savinov *et al.*, 2019; Pathak *et al.*, 2017)) only derive the novelty from local information, e.g., concatenation of few recent frames. Second, though prediction error has been widely adopted as an effective metric to infer novelty, most of the existing approaches develop novelty model upon short-term prediction error such as 1-step look-ahead. Such short-term prediction task might be an inadequate proxy for representing the novelty over state space, i.e., it might be too simple and thus result in inferior novelty scores.

Our key motivations are as follows. First, sequence-level novelty models are desired to reason over the partially observable states with greater efficiency. Second, the novelty model should consider longer-term prediction than self-prediction or 1-step look-ahead, to infer more meaningful novelty scores. Based on the above intuitions, this work proposes a new sequence-level novelty model for partially observable domains with the following two distinct properties. First, we introduce a dual-LSTM architecture to reason over a sequence of past transitions to construct the novelty model. Second, we infer the novelty of a state from the prediction error of open-loop multi-step forward dynamics prediction, which is crucial to derive high quality novelty scores.

## 2 METHODOLOGY

Partially Observable Markov Decision Process (POMDP) generalizes MDPs by learning under partial observability. Formally, a POMDP is defined as a tuple $\langle \mathcal{S}, \mathcal{A}, \mathcal{O}, \mathcal{T}, \mathcal{Z}, \mathcal{R} \rangle$, where $\mathcal{S}$, $\mathcal{A}$ and $\mathcal{O}$ are the spaces for the state, action and observation, respectively. The transition function $\mathcal{T}(s, a, s') = p(s'|s, a)$ specifies the probability for transiting to state $s'$ after taking action $a$ at state $s$. The observation function $\mathcal{Z}(s, a, o) = p(o|s, a)$ defines the probability of receiving observation $o$ after taking action $a$ at state $s$. The reward function $\mathcal{R}(s, a)$ defines the real-valued environment reward issued to the agent after taking action $a$ at state $s$. Under partial observability, the state space $\mathcal{S}$ is not accessible by the agent. Thus, the agent performs decision making by forming a *belief* state $b_t$ from its observation space $\mathcal{O}$, which integrates the information from the entire past history, i.e., $(o_0, a_0, o_1, a_1, ..., o_t, a_t)$. The goal of reinforcement learning is to optimize a policy $\pi(b_t)$ that outputs an action distribution given each *belief* state $b_t$, with the objective of maximizing the discounted cumulative rewards collected from each episode, i.e., $\sum_{t=0}^{\infty} \gamma^t r_t$, where $\gamma \in (0, 1]$ is a real-valued discount factor.

### 2.1 INTRINSIC EXPLORATION FRAMEWORK

We now describe our proposed sequence-level intrinsic novelty model for partially observable domains with high-dimensional inputs (i.e., images). Our primary focuses are the tasks where the external rewards $r_t$ are sparse, i.e., zero for most of the times. This motivates us to engage a novelty function to infer the novelty over the state space and assign reward bonus to encourage exploration.

The novelty function is derived from a forward-inverse dynamics model. Figure 1 depicts a high-level overview of our proposed sequence-level novelty computation. To infer the novelty of a state at time $t$, we perform reasoning over a sequence of transitions with length $L + H$. Intuitively, we use a sequence of $H$ consequent observation frames together with a sequence of actions with length $L$ which are taken following the observation sequence, to predict the forward dynamics. As such, the novelty model performs open-loop multi-step forward prediction. By setting the length of the

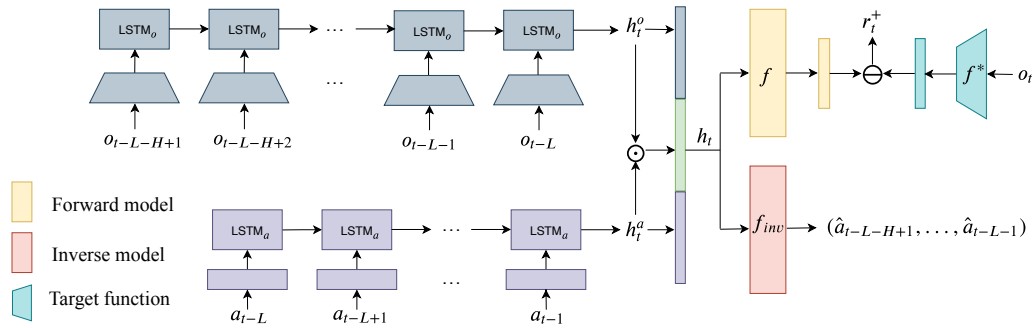

Figure 2: Dual-LSTM architecture for the proposed sequence-level intrinsic model. Overall, the forward model employs an observation sequence and an action sequence as input to predict the forward dynamics. The prediction target for forward model is computed from a target function $f^*(\cdot)$. An inverse dynamics model is employed to let the latent features $h_t$ encode more transition information.

action sequence, i.e., $L$, our proposed paradigm could lead to forward dynamics prediction tasks with varying difficulty.

To process the input sequences, we propose a dual-LSTM architecture as shown in Figure 2. Overall, each raw observation and action data are first projected by their corresponding embedding modules. Then LSTM modules are adopted over the sequences of observation/action embeddings to derive the sequential observation/action features. Then the sequential observation/action features are synthesized in a specific form of $h_t$, which serves as the latent representation for the past transitions at time $t$ and is employed as input to predict forward dynamics $f(h_t)$. The error of the forward dynamics prediction is used to estimate the novelty $r_t^+$ of the state at time $t$. Furthermore, to make the latent features over the past transitions more informative, we also incorporate an inverse dynamics prediction model $f_{inv}$ to predict the action distributions. Overall, the proposed dual-LSTM architecture enables us to perform sequence-level reasoning and inferring novelty from the multi-step forward prediction.

## 2.2 SEQUENCE ENCODING WITH DUAL-LSTM ARCHITECTURE

The sequence encoding module accepts a sequence of observations with length $H$ and a sequence of actions with length $L$ as input. Formally, we denote the observation sequence and action sequence by $\mathbf{O}_t = o_{t-L-H-1:t-L-1}$ and $\mathbf{A}_t = a_{t-L-1:t-1}$, respectively. Specifically, each observation $o_t$ is represented as a 3D image frame with width $m$, height $n$ and channel $c$, i.e., $o_t \in \mathbb{R}^{m \times n \times c}$. Each action is modeled as a 1-hot encoding vector $a_t \in \mathbb{R}^{|A|}$, where $|A|$ denotes the size of the action space.

Given the sequences $\mathbf{O}_t$ and $\mathbf{A}_t$, the sequence encoding module first adopts an embedding module $f_e(\cdot)$ parameterized by $\theta_E = \{\theta_{E_o}, \theta_{E_a}\}$ to process the observation sequence and the action sequence as follows,

$$\phi_t^{\mathbf{O}} = f_e(\mathbf{O}_t; \theta_{E_o}) \text{ and } \phi_t^{\mathbf{A}} = f_e(\mathbf{A}_t; \theta_{E_a}), \tag{1}$$

where $\theta_{E_o}$ and $\theta_{E_a}$ denote the parameters for the observation embedding function and the action embedding function, respectively. Next, LSTM encoders are applied to the output of the observation/action embedding modules as follows,

$$[h_t^o, c_t^o] = \text{LSTM}_o\left(\phi_t^{\mathbf{O}}, h_{t-1}^o, c_{t-1}^o\right) \text{ and } [h_t^a, c_t^a] = \text{LSTM}_a\left(\phi_t^{\mathbf{A}}, h_{t-1}^a, c_{t-1}^a\right), \tag{2}$$

where $h_t^o \in \mathbb{R}^l$ and $h_t^a \in \mathbb{R}^l$ represent the latent features encoded from the observation sequence and action sequence. For simplicity, we assume $h_t^o$ and $h_t^a$ have the same dimensionality. $c_t^o$ and $c_t^a$ denote the cell output for the two LSTM modules.

Next, the sequence features for the observation/action $h_t^o$ and $h_t^a$ are synthesized to derive latent features $h_t$ which describe the past transitions. Intuitively, the form of $h_t$ is proposed as follows:

$$h_t^{itr} = h_t^o \odot h_t^a \text{ and } h_t = [h_t^o, h_t^a, h_t^{itr}]. \tag{3}$$

To compute $h_t$, an multiplicative interaction is first performed over $h_t^o$ and $h_t^a$, which results in $h_t^{itr}$ and $\odot$ denotes element-wise multiplication. Then $h_t$ is derived by concatenating the multiplicative

interaction feature $h_t^{itr}$ with the latent representations for the observation and action sequences, i.e., $h_t^o$ and $h_t^a$. The reason for generating $h_t$ in this way is that the prediction task over the partial observation $o_t$ is related to both the local information conveyed in the two sequences themselves (i.e., $h_t^o$ and $h_t^a$), as well as the collaborative information derived via interacting the two sequence features in a form. The reason for performing multiplicative interaction is that the advancement of such operation in synthesizing different types of features has been validated in prior works (Oh *et al.*, 2015; Ma *et al.*, 2019). We demonstrate that generating $h_t$ in the proposed form is effective and crucial to derive a desirable policy learning performance in the ablation study (Figure 7c) of the experiment section.

## 2.3 Computing Novelty

To compute the novelty, the latent features $h_t$ are first employed as input by a feedforward prediction function to predict the forward dynamics:

$$\hat{\psi}_t = f(h_t; \theta_F) \ \text{ and } \ \psi_t^* = f_*(o_t), \tag{4}$$

where $f(\cdot)$ is the forward prediction function parameterized by $\theta_F$, and $\hat{\psi}_t$ denotes the prediction output. We use $\psi_t^*$ to denote the prediction target, which is computed from some target function $f_*(\cdot)$. Within the proposed novelty framework, the target function $f_*(\cdot)$ could be derived in various forms, where the common choices include the representation of $o_t$ at its original feature space, i.e., image pixels, and the learned embedding of $o_t$, i.e., $f_e(\cdot; \theta_{E_o})$. Apart from the conventional choices, in this work, we employ a target function computed from a random network distillation model (Burda *et al.*, 2019). Thus, $f_*(\cdot)$ is represented by a fixed and randomly initialized target network. Intuitively, it forms a random mapping from each input observation to a point in a $k$-dimensional space, i.e., $f_* : \mathbb{R}^{m \times n \times c} \to \mathbb{R}^k$. Hence the forward dynamics model is trained to distill the randomly drawn function from the prior. The prediction error inferred from such a model is related to the uncertainty quantification in predicting some constant zero function (Osband *et al.*, 2018).

The novelty of a state is inferred from the uncertainty evaluated as the MSE loss for the forward model. Formally, at step $t$, a novelty score or reward bonus is computed in the following form:

$$r^+(\mathbf{O}_t, \mathbf{A}_t) = \frac{\beta}{2} ||\psi_t^* - \hat{\psi}_t||_2^2, \tag{5}$$

where $\beta \geq 0$ is a hyperparameter to scale the reward bonus. The reward bonus is issued to the agent in a step-wise manner. During the policy learning process, the agent maximizes the sum over the external rewards and the intrinsic rewards derived from the novelty model. Therefore, the overall reward term to be maximized as will be shown in (8) is computed as $r_t = r_t^e + r_t^+$, where $r_t^e$ denotes the external rewards from the environment.

## 2.4 Loss Functions for Training

The training of the forward dynamics model is formulated as a regression problem. The loss for optimizing the forward dynamics model is defined as follows:

$$\mathcal{L}_F(\psi_t^*, \hat{\psi}_t) = \frac{1}{2} ||\psi_t^* - \hat{\psi}_t||_2^2. \tag{6}$$

We additionally incorporate an inverse dynamics model $f_{inv}$ over the latent features $h_t$ to make them encode more abundant transition information. Given the observation sequence $\mathbf{O}_t$ with length $H$, the inverse model is trained to predict the $H - 1$ actions taken between the observations. Thus, the inverse model is defined as:

$$f_{inv}(h_t; \theta_I) = \prod_{i=1}^{H-1} p(\hat{a}_{t-L-i}), \tag{7}$$

where $f_{inv}(\cdot)$ denotes the inverse function parameterized by $\theta_I$, and $p(\hat{a}_{t-L-i})$ denotes the action distribution output for time step $t - L - i$. The inverse model is trained by minimizing a standard cross-entropy loss.

Overall, the forward loss and inverse loss are jointly optimized with the reinforcement learning objective. Moreover, the parameters for the observation embedding module $\theta_{E_o}$ could be shared

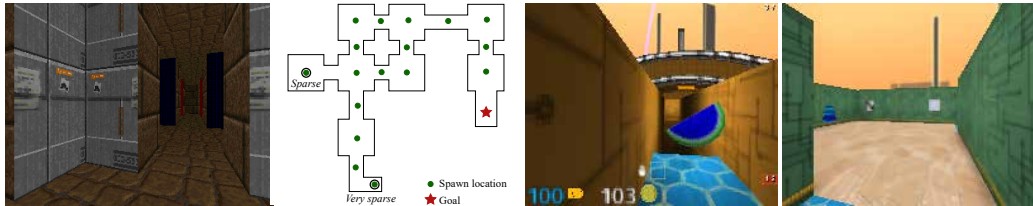

Figure 3: The 3D navigation task domains adopted for empirical evaluation: (1) an example of partial observation frame from ViZDoom task; (2) the spawn/goal location settings for ViZDoom tasks; (3/4) an example of partial observation frame from the apple-distractions/goal-exploration task in DeepMind Lab.

with the policy model. In summary, the compound objective function for deriving the intrinsically motivated reinforcement learning policy becomes:

$$\min_{\theta_E, \theta_F, \theta_I, \theta_\pi} \lambda \mathcal{L}_F(\psi_t^*, \hat{\psi}_t) + \frac{(1-\lambda)}{H-1} \sum_{i=1}^{H-1} \mathcal{L}_I(\hat{a}_{t-L-i}, a_{t-L-i}) - \eta \mathbb{E}_{\pi(\phi_t^o; \theta_\pi)} \left[ \sum_t r_t \right], \quad (8)$$

where $\theta_E$, $\theta_F$ and $\theta_I$ are the parameters for the novelty model, $\theta_\pi$ are the parameters for the policy model, $\mathcal{L}_I(\cdot)$ is the cross-entropy loss for the inverse model, $0 \le \lambda \le 1$ is a weight to balance the loss for the forward and inverse models, and $\eta \ge 0$ is the weight for maximizing the cumulative reward.

## 3 EXPERIMENTS

### 3.1 EXPERIMENTAL SETUP

**Task Domains** For empirical evaluation, we adopt three 3D navigation tasks with first-person view: 1) '*DoomMyWayHome-v0*' from ViZDoom (Kempka *et al.*, 2016); 2) '*Stairway to Melon*' from DeepMind Lab (Beattie *et al.*, 2016); 3) '*Explore Goal Locations*' from DeepMind Lab. The experiments in '*DoomMyWayHome-v0*' allow us to test the algorithms in scenarios with varying degrees of reward sparsity. The experiments in '*Stairway to Melon*' allow us to test the algorithms in scenarios with reward distractions. The experiments in '*Explore Goal Locations*' allow us to test the algorithms in scenarios with procedurally generated maze layout and random goal locations.

**Baseline Methods** For fare comparison, we adopt 'LSTM-A3C' as the RL algorithm for all the methods. In the experiments, we compare with the vanilla 'LSTM-A3C' as well as the following intrinsic exploration baselines: 1) the Intrinsic Curiosity Module (Pathak *et al.*, 2017), denoted as 'ICM'; 2) Episodic Curiosity through reachability (Savinov *et al.*, 2019), denoted as 'EC'; 3) the Random Network Distillation model, denoted as 'RND'. Our proposed Sequence-level Intrinsic exploration Module is denoted as 'SIM'. Our method adopt observation length 10 and action length 3 consistently for all domains. All the intrinsic exploration baselines adopt non-sequential inputs. The baseline 'EC' is a memory-based algorithm. We shift the corresponding learning curves by the budgets of pretraining frames (i.e., 0.6M) in the results to be presented, following the original paper (Savinov *et al.*, 2019). We present the implementation details for all the compared methods in Appendix A.

### 3.2 EVALUATION WITH VARYING REWARD SPARSITY

Our first empirical domain is a navigation task in the '*DoomMyWayHome-v0*' scenario from ViZDoom. The task consists of a static maze layout and a fixed goal location. At the start of each episode, the agent spawns from one of the 17 spawning locations, as shown in Figure 3. In this domain, we adopt three different setups with varying degree of reward sparsity, i.e., *dense*, *sparse*, and *very sparse*. Under the *dense* setting, the agent spawns at one randomly selected location from the 17 locations and it is relatively easy to succeed in navigation. Under the *sparse* and *very sparse* settings, the agent spawns at a fixed location far away from the goal. The environment issues a positive reward of +1 to the agent when reaching the goal. Otherwise, the rewards are 0. The episode terminates when the agent reaches the goal location or the episode length exceeds the time limit of 525 4-repeated steps.

We show the training curves measured in terms of navigation success ratio in Figure 4. The results from Figure 4 depicts that as the rewards go sparser, the navigation would become more challenging.

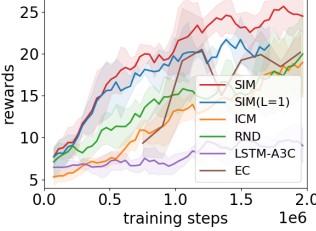

Figure 4: Learning curves measured in terms of the navigation success ratio in ViZDoom. The figures are ordered as: 1) *dense*; 2) *sparse*; 3) *very sparse*. We run each method for 6 times.

The vanilla 'LSTM-A3C' algorithm could not progress at all under the *sparse* and *very sparse* settings. 'ICM' could not reach 100% success ratio under the *sparse* and *very sparse* settings, and so does 'EC' under the *very sparse* setting. Our proposed method consistently achieves 100% success ratio across all the tasks with varying reward sparsity. The detailed convergence scores are shown in Table 1. We also present the results measured in terms of average episode length in Appendix B.3.

Our proposed solution also demonstrates significant advantage in terms of convergence speed. Though the reward sparsity varies, our method could quickly reach 100% success ratio in all the scenarios. However, the convergence speeds of 'ICM', 'EC' and 'RND' apparently degrade with sparser rewards. Also, we notice that the memory-based method (i.e., 'EC') takes much longer time to converge compared to the prediction-error based baselines 'RND' and 'SIM'. That is, the learning curves for those prediction-error based methods go up with a much steeper ratio compared to the memory-based method. The reason might come from the memory that 'EC' keeps and updates at run-time to infer the novelty. The novelty score assigned for each state might be unstable due to the run-time update to memory. Moreover, 'EC' requires to pre-train the comparator module in some task domains such as ViZDoom, whereas our method, as well as 'ICM' and 'RND', does not require pre-training. Overall, our proposed method could converge to 100% success ratio on average 3.1x as fast as 'ICM' and 2.0x compared to 'RND'. We present some detailed convergence statistics in Appendix B.4.

## 3.3 EVALUATION WITH VARYING MAZE LAYOUT AND GOAL LOCATION

Our second empirical evaluation engages a more dynamic navigation task with procedurally generated maze layout and randomly chosen goal locations. We adopt the 'Explore Goal Locations' level script from DeepMind Lab. At the start of each episode, the agent spawns at a random location and searches for a randomly defined goal location within the time limit of 1350 4-repeated steps. Each time the agent reaches the goal, it receives a reward of +10 and is spawned into another random location to search for the next random goal. The maze layout is procedurally generated at the start of each episode. This domain challenges the algorithms to derive general navigation behavior instead of relying on remembering the past trajectories.

We show the results with an environment interaction budget of 2M 4-repeated steps in Figure 5. We exempt the baseline 'EC' in this task, because the pretraining of 'EC' consumes 0.6M interaction budgets, which makes it less feasible for the current task. As a result, the method without intrinsic novelty model could only converge to an inferior performance around 10. Our proposed method could score > 20 with less than 1M training steps, whereas 'ICM' and 'RND' take almost 2M steps

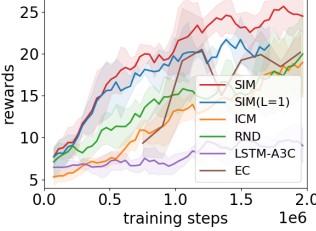

Figure 5: Learning curves for the procedurally generated goal searching task in Deep-Mind Lab. We run each method for 5 times.

|  | dense | sparse | very sparse |
|---|---|---|---|
| LSTM-A3C | 100% | 0.0% | 0.0% |
| ICM | 100% | 66.7% | 68.6% |
| EC | 100% | 100% | 75.5% |
| RND | 100% | 100% | 100% |
| SIM | 100% | 100% | 100% |

Table 1: Performance scores for the three task settings in ViZDoom evaluated over 6 independent runs. Overall, only our approach and 'RND' could converge to 100% under all the settings. Our method on average converges 2.0x as fast as 'RND' and 3.1x as fast as 'ICM' in ViZDoom domains.

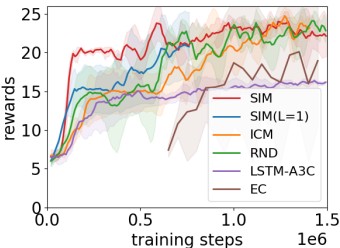 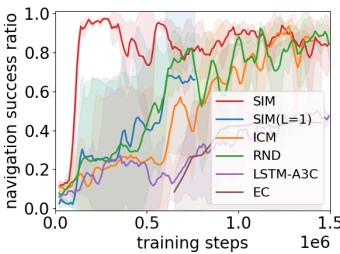

Figure 6: Learning curves for 'Stairway to Melon' task in DeepMind Lab. **Left**: cumulative episode reward; **Right**: navigation success ratio. We run each method for 5 times.

to score above 20. This demonstrates that our proposed algorithm could progress at a much faster speed compared to all the baselines under the procedurally generated maze setting.

## 3.4 EVALUATION WITH REWARD DISTRACTIONS

Our third empirical evaluation engages a cognitively complex task with reward distraction. We adopt the '*Stairway to Melon*' level script from DeepMind Lab. In this task, the agent can follow either two corridors: one of them leads to a dead end, but has multiple apples along the way, collecting which the agent would receive a small positive reward of $+1$; the other corridor consists of one lemon which gives the agent a negative reward of $-1$, but after passing the lemon, there are stairs that lead to the navigation goal location upstairs indicated by a melon. Collecting the melon makes the agent succeed in navigation and receive a reward of $+20$. The episode terminates when the agent reaches the goal location or the episode length exceeds the time limit of 525 4-repeated steps.

The results are shown in Figure 6. We show both the cumulative episode reward and the success ratio for navigation. Due to the reward distractions, the learning curves for each approach demonstrate instability with ubiquitous glitches. The vanilla 'LSTM-A3C' could only converge to an inferior navigation success ratio of $< 50\%$, and all the other baselines progress slowly. Notably, our proposed method could fast grasp the navigation behavior under the reward distraction scenario, i.e., surpassing the standard of $> 80\%$ with less than 0.2M environment interactions, which is at least 3x as fast as the compared baselines.

## 3.5 ABLATION STUDY

In this section, we present the results for an ablation study under the *very sparse* task in ViZDoom.

**Impact of multi-step prediction:** We demonstrate that performing multi-step prediction could be beneficial for policy training. In Figure 4 (c), we have shown the comparison results with self-prediction baseline 'RND' and one-step prediction baseline 'ICM', both of which are feed-forward models. In this section, we show the results by comparing with sequence-level one-step prediction baselines adapted from our proposed model. From the results shown in Figure 7a, we notice that performing 3-step forward prediction would result in apparently better convergence than the 'L1' variants. Expanding the scale of prediction difficulty by incorporating longer-term forward prediction would be beneficial to derive high-quality novelty scores than one-step models.

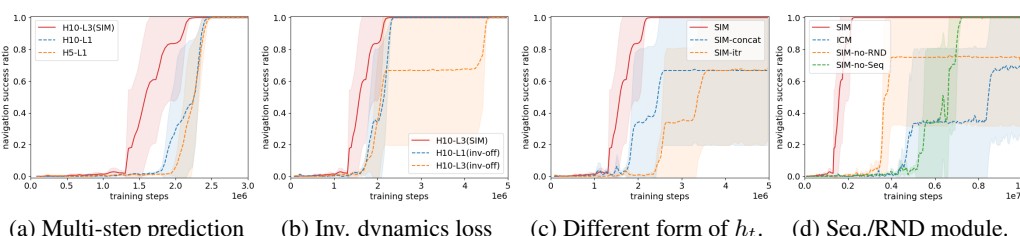

(a) Multi-step prediction  (b) Inv. dynamics loss  (c) Different form of $h_t$.  (d) Seq./RND module.

Figure 7: Results of ablation study in the *very sparse* task of ViZDoom.

**Impact of inverse dynamics loss:** We also investigate the impact of shaping the latent representation $h_t$ by incorporating the inverse dynamics loss. To this end, we show the performance of our proposed model when the inverse dynamics is turned off in Figure 7b. When performing short-term prediction, such as one-step look-ahead, the effect of inverse dynamics might not be very significant. However, when considering longer term prediction, utilizing inverse dynamics loss could efficiently stabilize the training and help to shape the latent representation to be more meaningful (i.e., performance of H10-L3 (inv-off) is much worse than H10-L1 (inv-off)).

**Impact of $h_t$:** We demonstrate that modeling $h_t$ in the proposed form of (3) is efficient by comparing our method with the following two baseline models of $h_t$: 1) only using the interactive features $h_t^{itr}$, denoted by 'SIM-itr', and 2) only using the concatenation of $h_t^o$ and $h_t^a$, denoted by 'SIM-concat'. From the results shown in Figure 7c, we find that both baseline methods converge to inferior performance standard, i.e., the algorithm fail occasionally so that the averaged curve could not converge to $100\%$ success ratio. When using $h_t$ in the proposed form, the algorithm could consistently converge to $100\%$ success ratio. This demonstrates that modeling $h_t$ in our proposed form is crucial for deriving a desired policy learning performance.

**Impact of the sequence/RND module:** Lastly, we testify the efficiency of the two critical parts for our solution: 1) the sequence embedding module with dual-LSTM; 2) the RND module to compute the prediction target. To this end, we create the following two baselines: 1) using a feedforward model together with RND, denoted by 'SIM-no-Seq', and 2) training the sequence embedding model with the target computed from the embedding function $f_e(\cdot; \theta_{E_o})$ instead of RND, denoted by 'SIM-no-RND'. The results are shown in Figure 7d. 'SIM-no-Seq' could outperform the 'ICM' baseline, which indicates that using random network distillation to form the target could be more efficient in representing the novelty of state than using the learned embedding function. Also, 'SIM-no-RND' could converge much faster than 'ICM', which indicates that using the sequence-level modeling of novelty is more efficient than using flat concatenation of frames. Overall, this study shows that using the sequence embedding model together with the RND prediction target is critical for deriving desirable performance.

## 4 RELATED WORK

Curiosity-driven exploration has been studied extensively in the reinforcement learning literature. We refer the readers to (Oudeyer and Kaplan, 2009; Oudeyer et al., 2007) for an overview. In recent years, research on intrinsic exploration for deep reinforcement learning develops the novelty or curiosity model based on various factors, such as counts (Tang et al., 2017; Choi et al., 2019), pseudo-counts (Bellemare et al., 2016; Ostrovski et al., 2017), prediction-error (Achiam and Sastry, 2017; Stadie et al., 2015) and information gain (Houthooft et al., 2016; Nikolov et al., 2019).

A prominent line of approaches for intrinsic exploration under partially observable settings fall under the prediction-error-based approaches. Pathak et al. (2017) propose a forward-backward dynamics model trained with self-supervision, and use the prediction loss of the forward model to infer the state novelty. Oh and Cavallaro (2019) incorporate a triplet ranking function to push the prediction output of the forward model to be far from some alternative prediction output computed with wrong action inputs. Apart from those prediction-error-based approaches, recently, Savinov et al. (2019) propose a memory-based approach which forms a memory of novel states and trains a comparator network to model the reachability between states to assign state novelty. While all the above mentioned approaches adopt visual inputs modeled as flat concatenation of frames, we model the sequence-level novelty from past transition sequence. Compared to the recent works that adopt sequence-level modeling in policy training (Chiappa et al., 2017; Ke et al., 2019), they mainly consider sequence-level policy, or construct dynamics models that are autoregressive. In our work, we propose a dual-LSTM architecture that tackles open-loop multi-step dynamics prediction.

## 5 CONCLUSION

In this paper, we tackle the challenge of improving policy training in sparse rewarded partially observable domains. We propose a sequence-level novelty model, and we demonstrate the benefit of such a model through various experimental domains, including tasks with partially observability as well as fully observable tasks. In the future, we want to explore the possibility of adapting our proposed solution to derive modularized and transferable novelty model among related task domains.

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

## APPENDICES

This supplementary material is organized as follows. First, we describe the implementation details for our proposed model and the baseline models. Then we demonstrate the additional experiment results for the ViZDoom domains. Besides, we show the task settings and hyperparameter settings to replicate the results shown for the 'Stairway to Melon' and 'Explore Go Locations Small' tasks in DeepMind Lab. Finally, we present the results on Atari experiments.

## A  IMPLEMENTATION DETAILS

### A.1  SIM

The novelty model employs a sequence of observation frames with length $H$ and a sequence of actions with length $L$ as its input. Specifically, each observation is modeled as RGB images of size $42 \times 42 \times 3$. Each action is modeled as one-hot vector with size $|\mathcal{A}|$, where $\mathcal{A}$ is the discrete action space used for each domain. The observation embedding module $f_e(\cdot; \theta_{E_o})$ consists of two *conv* layers with 16 and 32 filters respectively. Both layers use kernel size of $3 \times 3$ and stride of 2. The convolution layers are followed by a *fc* layer of size 256. The action embedding layer is a *fc* layer of size 128. Then the state and action embedding outputs are fed to two LSTMs with latent size 256. The synthesized latent feature $h_t$ has a size of $256 \times 3$. $h_t$ is fed to a *fc* layer of size 64 to construct the forward model (to predict RND target) and another *fc* layer of size $(H - 1) * |A|$ followed by a *softmax* layer to construct the inverse model. The inverse model predicts the previously taken actions with length $H - 1$. The target function $f_*(\cdot)$ is modeled by random network distillation function, for which we employ the same setting as introduced in Section A.3.

### A.2  ICM

The ICM model takes an observation and an action as its input to predict the embedding computed for the observation of the next state. The observation is modeled as 4 consequent gray-scale frames with size $42 \times 42 \times 4$. The action input is modeled as an one-hot vector with size $\mathcal{A}$. We adopt the identical ICM implementation as the open-source code released for the original paper. Specifically, the observation embedding model consists of 4 consequent *conv* layers, with filter size of 32, kernel size of $3 \times 3$ and stride of 2 for each. The output of the observation embedding is a vector of size 288. The observation embedding is concatenated with the one-hot action to form a synthesized feature, which is fed to a *fc* layer of size 256 followed by another *fc* of size 288 to form the forward dynamics model.

### A.3  RND

The RND mdoel takes an RGB image of size $42 \times 42 \times 3$ as its input to predict the randomly projected target for that RGB image. Specifically, the target function consists of 3 *conv* layers with 32 filters for each, kernel sizes of 8, 4 and 3, and strides of 4, 2 and 1, respectively. The outputs of the last *conv* layer is followed by a *fc* layer of size 64 to compute the prediction target. For all the layers, the weights are initialized by orthogonal initializer with scale of $\sqrt{2}$ and the biases for all the layers are initialized by 0-initialization. The prediction model consists of 4 consequent *conv* layers, with filter size of 32, kernel size of $3 \times 3$ and stride of 2 for each. The output of last *conv* layer is followed by a *fc* layer of size 64 to output the prediction.

### A.4  EC

The comparitor network for 'EC' is implemented as Resnet-18 with 512 outputs. The concatenated features for a pair of compared frames are fed to a *fc* layer of size 512, which is followed by a *fc* layer of size 2 and a *softmax* function to compute the classification probability. We adopt the identical implementation from the open-source code of the original paper.

## A.5 LSTM-A3C

At each step, the 'LSTM-A3C' policy model takes an RGB image of size $42 \times 42 \times 3$ as its input. The image is first processed by an embedding module, which consists of two *conv* layers with 16 and 32 filters, kernel size of $3 \times 3$, and stride of 2 for each. Then the output of the last *conv* layer is fed to a *fc* layer of size 256 to form the input to the LSTM module. We adopt single LSTM layer with a latent size of 256. The output of the LSTM module is fed to two brunches: one followed by a *fc* layer of size 1 to predict the value, another followed by a *fc* layer of size $|\mathcal{A}|$ and a *softmax* layer to predict the action probability. To train the 'LSTM-A3C', we use ADAM optimizer with learning rate of $1e-4$ for all the task domains. For ViZDoom tasks, we adopt 16 asynchronous actors and for DeepMind Lab tasks, we adopt 8 actors. The value loss is weighted by $0.5$ and the entropy regularization weight is 0.01.

# B   VIZDOOM TASKS

## B.1   TASK SETTINGS

We adopt the 'DoomMyWayHome-v0' scenario for the navigation tasks in ViZDoom. Specifically, the tasks are static goal reaching tasks with a static maze layout. The three task settings differ in terms of the agent's spawn location. Each episode lasts for 525 4-repeated simulator steps. The episode terminates when the agent reaches the static goal location. We adopt a discrete action space with size of 4: {*move-left, move-right, forward, no-op*}.

## B.2   HYPERPARAMETER SETTINGS

|  | LSTM-A3C | ICM | RND | EC | SIM |
|---|---|---|---|---|---|
| learning rate | 1e-4 | 1e-4 | 1e-4 | 1e-4 | 1e-4 |
| task reward scale | 1.0 | 1.0 | 1.0 | 1.0 | 1.0 |
| bonus reward scale | - | 0.01 | 0.01 | 0.01 | 0.001 |
| ICM forward inverse ratio ($\lambda$) | - | 0.2 | - | - | 0.2 |
| RL loss coefficient ($\eta$) | - | 0.1 | 0.1 | - | 0.1 |

## B.3   EVALUATION RESULT IN TERMS OF EPISODIC LENGTH

In this static goal reaching domain, the agent with better navigation policy would result in shorter path to reach the goal. Therefore, we present the results evaluated in terms of average episodic length for each algorithm under the three task settings. The results are shown in Figure 8.

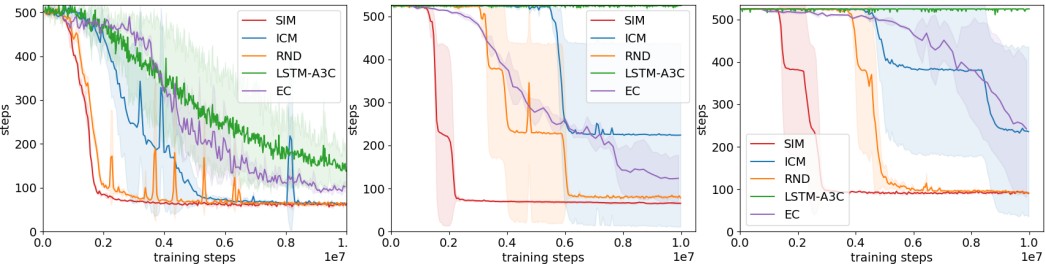

Figure 8: average steps of episode: (1) Vizdoom dense (2) Vizdoom sparse (3): Vizdoom very sparse

We observe that the episodic lengths decrease at different ratios for the compared algorithms. Notably, our algorithm decreases at a much faster ratio compared to all the baselines. Meanwhile, the episodic length for both 'SIM' and 'RND' would converge to a better standard than the other methods.

### B.4 EVALUATION RESULT IN TERMS OF CONVERGENCE TIME

To qualitatively evaluate the advantage of our algorithm in terms of accelerating the policy learning convergence, we measure the average convergence time for each algorithm across the ViZDoom tasks. The resuls are shown in Table 2. For each algorithm, we present the approximated environment steps required for the algorithm to reach the convergence standard.

|             | LSTM-A3C | ICM   | RND   | EC    | SIM (ours) |
|-------------|----------|-------|-------|-------|------------|
| *dense*     | 7.13m    | 3.50m | 1.86m | >10m  | 1.42m      |
| *sparse*    | >10m     | 6.01m | 4.51m | 6.45m | 1.82m      |
| *very sparse* | >10m   | 6.93m | 4.55m | >10m  | 1.97m      |

Table 2: The approximated environment steps taken by each algorithm to reach its convergence standard under each task setting. Notably, our proposed algorithm could achieve an average speed up of 3.1x compared to 'ICM', and 2.0x compared to 'RND'.

### B.5 A STUDY ON THE IMPACT OF SCALING FACTOR ($\beta$)

In response to the reviewer's comments, we provide a study on the impact of scaling factor on the tasks with both dense reward and sparse reward setting. To this end, we present the result on the following two settings from ViZDoom: *dense* and *verySparse*. Specifically, we testify the effect of policy training by investigating on the scaling factor from the following set: $\{0.01, 0.001, 0.0001\}$, where 0.001 is our recommended setting that works well across the ViZDoom domains and 0.01 and 0.0001 are the apparently large/small settings.

First, we present the policy training rewards under each scaling factor setting in Figure 9. In the *dense* reward scenario, the policy could learn with both scaling factors of 0.001 and 0.0001. However, when we set it to be 0.01, it would result in bias in behavior and lead to inferior task performance. Thus, we could conclude that the *dense* reward tasks could still benefit from such reward shaping with proper setting of the scaling factor. For the *verySparse* setting, we observe that when the scaling factor is too small, its effect is very minimal and are not suffice to let the policy learning progress. Also, when it is set to be very large, it causes biased behavior and let the policy fail to learn proper behavior. But generally, the suggested value 0.001 works well on all the cases.

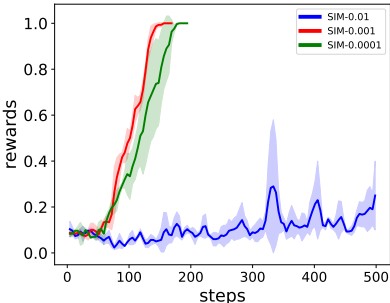 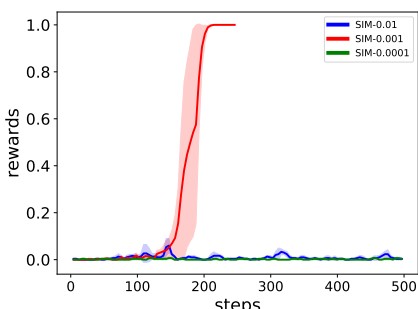

Figure 9: Comparing different scaling factor setting on the following two ViZDoom scenarios: *dense* (left) and *very sparse* (right). Results are presented in terms of task rewards.

We also present the scale of the intrinsic reward in Figure 10. Each data point corresonds to the cumulative episodic intrinsic reward evaluated at the given training time. Overall, the intrinsic reward decreases as the training progresses, which means that our dynamics model is getting trained properly. The result reveals that if we set the scaling factor to be too large, it would result in large intrinsic rewards and lead to inferior policy training performance.

In Figure 10 (right), we observe that with the scaling factor 0.001, there is a peak between 1m-2m steps, where the agent intensively explore novelty states to learn meaningful behavior policy. After

2m steps, the model has converged to optimal behavior and the agent no longer receive intrinsic rewards at large scale any more.

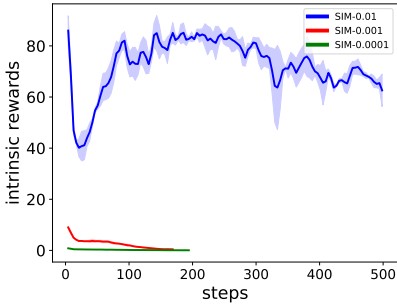 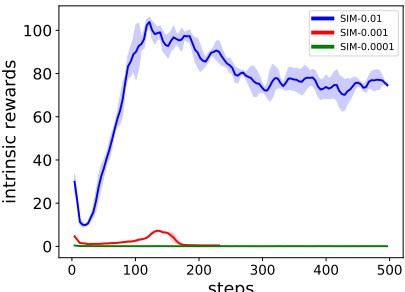

Figure 10: Comparing different scaling factor setting on the following two ViZDoom scenarios: *dense* (left) and *very sparse* (right). We present the episodic cumulative intrinsic rewards.

### B.6 A STUDY ON THE IMPACT OF MULTI-STEP SIZE($L$)

In response to the reviewer's comments, we present a study on performing multi-step forward prediction with the size of forward prediction step (i.e., $L$) selected from the following set $\{1, 2, 3, 4, 5\}$. From the results shown in Figure 11 , we notice that performing multi-step prediction with our suggested $L = 3$ would have apparent advantage over $L = 1$ and $L = 2$. There is no apparent advantage for $L = 4$ and $L = 5$ over $L = 3$. Therefore, we suggest $L = 3$ as a recommended hyperparameter setting.

The phenomena that overlength prediction would not further benefit policy learning could be explained as follows. The open-loop forward prediction we introduce could efficiently scale up the prediction difficulty, i.e., the larget $L$, the more difficult to predict. When the prediction is too difficult, the model leads to less favorable intrinsic reward bonus, among which the difference between prediction errors for novel states would be less distinguishable from those common states. Therefore, performing multi-step forward prediction with moderate $L$ would be ideal.

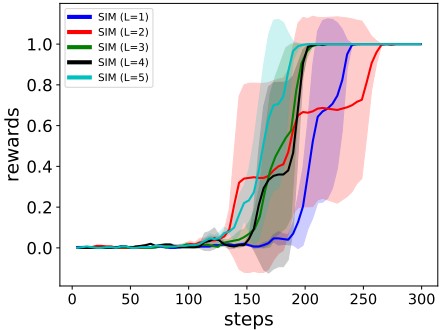

Figure 11: Experiment result on performing multi-step forward prediction evaluated on *verySparse* ViZDoom task.

# C  'STAIRWAY TO LEMON' TASK

## C.1  TASK SETTINGS

We adopt the 'Stairway to Lemon' level script from DeepMind Lab to testify the algorithms under static goal reaching task with reward distraction. This task engages a static maze and a static navigation target location. Each episode terminates when the agent reaches the navigation location or the episodie length exceeds a predefined time out of 2100 4-repeated simulator steps. We adopt a discrete action space with size of 3: {*move-left*, *move-right*, *forward*}.

## C.2  HYPERPARAMETER SETTINGS

|                                     | LSTM-A3C | ICM  | RND  | EC   | SIM   |
|-------------------------------------|----------|------|------|------|-------|
| learning rate                       | 1e-4     | 1e-4 | 1e-4 | 1e-4 | 1e-4  |
| task reward scale                   | 0.1      | 0.1  | 0.1  | 0.2  | 0.1   |
| bonus reward scale                  | -        | 0.05 | 0.1  | -    | 0.001 |
| ICM forward inverse ratio ($\lambda$) | -      | 0.2  | -    | -    | 0.2   |
| RL loss coefficient ($\eta$)        | -        | 0.1  | 0.1  | -    | 0.9   |

## C.3  EVALUATION IN TERMS OF EPISODIC LENGTH

In this task domain, we observe that the reward distractions would result in significant instability to the learning process. We show the evaluation result measured in terms of average episodic length in Figure 12. The result shows that our method leads to significantly faster decrease in average episodic length compared to the baseline methods.

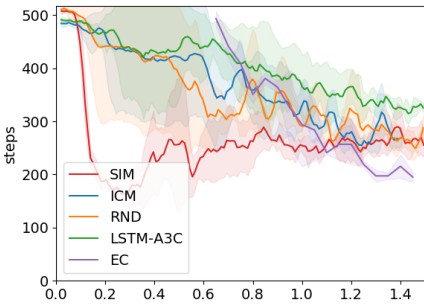

Figure 12: Average episodic length in the 'Stairway to melon' task from DeepMind Lab.

# D  'EXPLORE GOAL LOCATIONS SMALL' TASK

## D.1  TASK SETTINGS

We adopt the 'Explore Goal Locations Small' level script from DeepMind Lab to testify the algorithms under dynamic goal reaching settings. Specifically, this task engages an episodically generated maze with a randomly specified goal location. It challenges the learning algorithms to derive generalizable navigation behaviors that do not depend on remembering the static goal reaching paths. We terminate each episode when the agent exceeds a predefined time out of 1350 4-repeated simulator steps. Same to the previous task, we adopt a discrete action space with size of 3: {*move-left*, *move-right*, *forward*}.

|                                  | LSTM-A3C | ICM  | RND  | EC   | SIM   |
|----------------------------------|----------|------|------|------|-------|
| learning rate                    | 1e-4     | 1e-4 | 1e-4 | 1e-4 | 1e-4  |
| task reward scale                | 0.1      | 0.1  | 0.1  | 0.2  | 0.1   |
| bonus reward scale               | -        | 0.05 | 0.01 | -    | 0.001 |
| ICM forward inverse ratio ($\lambda$) | -   | 0.2  | -    | -    | 0.2   |
| RL loss coefficient ($\eta$)     | -        | 0.1  | 0.1  | -    | 0.1   |

## D.2 Hyperparameter Settings

# E Atari 2600 Tasks

## E.1 Hard Exploration Tasks

Though our exploration model is not designated for Atari tasks, we still get some results on the hard exploration Atari domains. By investigating on the performance of our model in Atari tasks, we could evaluate the generality of our model on less partially observable domains. At the same time, we could testify the scalability of our dynamics prediction model on tasks with relatively large action space.

In the experiment, we compare our method with the following two exploration baselines: 'ICM' and 'RND'. The algorithms are testified in the following two hard exloration Atari tasks: *ms-pacman*, which has 9 actions; *solaris*, which has 18 actions. We implement the exploration models on top of the IMPALA framework. Each compared method is run with identical environment settings.

The results are presented in Figure 13. In *ms-pacman*, our proposed method could progress much faster than the two baselines. Moreover, the model leads to much higher convergence performance than 'ICM' and 'RND'. In *solaris*, which is a challenging task in which each algorithm results in noisy learning curve with glitches, our method could achieve highest *best* score. Overall, the results in Atari domains reveal that our novelty model is promising to work in a variety of task domains with different degree of observability.

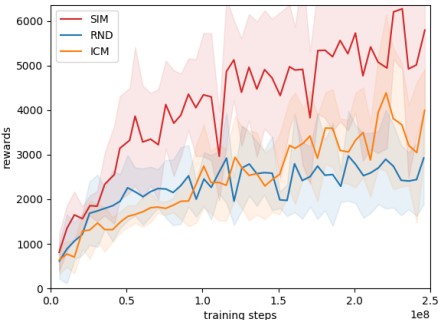 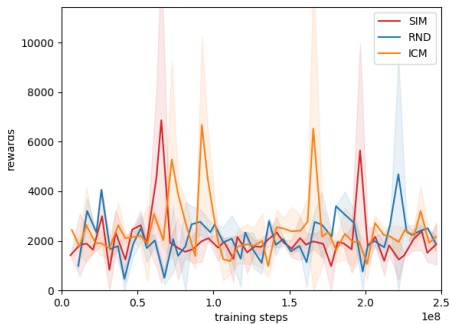

Figure 13: Learning curves for two hard exploration tasks in Atari 2600 domain. **Left**: ms-pacman (9 actions); **Right**: solaris (18 actions).

## E.2 Dense-rewarded Task with Large Action Space

In response to the reviewer's comments, we provide experiment result on another complex dense reward task domain. We take *Seaquest* from Atari 2600 as an example. Besides dense reward, the task consists of another appealing characteristic which is that it has a relatively large action space that consists of 18 control actions. Evaluating the method at this task domain could also help us to investigate whether our proposed forward dynamics-based reward shaping method could work on tasks with large action space. The experiment result is shown in Figure 14. The policy trained with our proposed method could progress significantly faster than RND and ICM. Moreover, our method also leads to better convergence reward compared to the baselines. It demonstrates that our proposed

method could benefit the dense-rewarded tasks as well. Moreover, it also shows the generality of our proposed solution to tackle a broad range of RL tasks, not being restricted to deal with only partially observable navigation tasks.

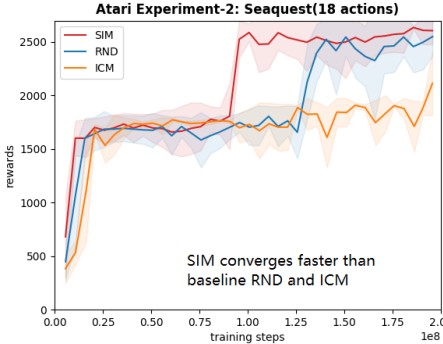

Figure 14: Learning curves for a dense-rewarded Atari 2600 task *Seaquest* which has a large action space that consists of 18 actions.

