# OpenReview forum: "Sequence-level Intrinsic Exploration Model for Partially Observable Domains"
_ICLR.cc/2020/Conference — Reject_

### Official Review · AnonReviewer2 · 2019-10-07
**Official Blind Review #2**

**Rating:** 6

**Review:**


The authors tackle the exploration problem by introducing SIM (Sequence-level Intrinsic exploration Module). In most existing literature, intrinsic motivation bonuses are scored based on individual states or transitions, and not over multi-step trajectories. SIM predicts novelty bonuses based on the prediction error of an open-loop forward dynamics model - the model consumes as input a sequence of observations (without paired actions) followed by a sequence of actions (without paired observations) to predict a feature vector associated with the next state. The error between this feature vector and the RND embedding of the true observation is used as a novelty bonus.

Overall, the paper is easy to follow and well-motivated. The main experimental results show a reasonable improvement over baselines (RND, ICM). While the model contains many moving components that may seem ad-hoc, the ablation studies show the beneficial effect of each of the individual modeling choices (specifically, using multi-step predictions, the auxiliary inverse dynamics loss, and RND embedding). It would be nice if experiments could be performed on a more popular benchmark such as Atari, but overall I think this is an interesting paper with a reasonable contribution.

For additional motivation, "Why is posterior sampling better than optimism for reinforcement learning" (Osband & Van Roy 2016) offers some justification on the downsides of modeling novelty bonuses for each state/action independently.

**Experience Assessment:**

I have published in this field for several years.

**Review Assessment: Checking Correctness Of Derivations And Theory:**

N/A

**Review Assessment: Checking Correctness Of Experiments:**

I carefully checked the experiments.

**Review Assessment: Thoroughness In Paper Reading:**

I read the paper thoroughly.

---

> ### Author Response · Authors · 2019-11-14
> **Author Response to Review #2**
>
> We thank the reviewer for providing such an informative feedback.
>
> Results on three Atari tasks are available in Appendix E. We consider posterior sampling as an appealing way for performing statistically efficient exploration. We are interested in investigating more on this direction in our future work.

---

### Official Review · AnonReviewer1 · 2019-10-22
**Official Blind Review #1**

**Rating:** 3

**Review:**

This paper extends the prediction-error based model by Pathak et. al., 2019 by learning a forward (and inverse) dynamics model for predicting a state feature multiple steps into the future (say, K-steps) given an open loop sequence of K actions as opposed to 1 step into the future, with the caveat that instead of using learnable state features, a random network is used for computing state features, similar to Random Network Distillation (RND) by Burda et. al., 2019. Also, their inverse dynamics models predicts the entire sequence of actions up to K steps. Experiments on VizDoom point-navigation tasks show that the proposed model does better than baselines as rewards get sparser. Ablations are provided to justify the choice of K in multi-step prediction, the choice of inverse dynamics and the choice of RND state features.

My decision is weak reject as:

1. The paper does a good job at clearly explaining their model, presenting results on relevant experiments and baseline comparisons for their model and justifying each modeling choice with ablations.

2. The novelty in their contribution is moderate - the idea of long-term future prediction is not new (e.g.: Ke et. al., 2019) (but using it for giving a curiosity bonus is new), the architecture choice is not significantly new.

3. I expected a more detailed ablation for the choice of K, given that “multi-step” has major emphasis in the paper, but Figure 7(a) only shows ablations for 3 vs 1 step predictions.


Elaborating on (3):
- My major concern is that the gap between 1-step prediction and 3-step prediction (in Figure 7 (a)) is not significant. Note that the version of their model with 1-step predictions does not completely reduce to Pathak et. al. 2019’s ICM model, as RND state features are used. I feel that it may be the case that the 1-step version of the proposed model is actually good enough to beat all the baselines and adding multiple steps gives marginal gains. This hypothesis needs to be verified by the authors with more experiments - I would like to see all the main experiments have an additional baseline of 1-step predictions.

- Only two values of K are tested - 1 and 3, what happens with larger values of K?

Other comments:
The motivation for using long-term predictions to “infer more meaningful novelty” is fine on it’s own but seems to conflict with the choice of random network (RND) state features. Random features imply that a random “hash” of the observations is being computed which has no reason to have similar features for two nearby states. If there is any small amount of noise in the state transitions, this would mean that predicting far into the future is practically impossible given that the random feature of slightly incorrect states will be very different. Can the authors give reasons/motivations as to why such a model would work in the case of stochastic transitions and K is large or will it be brittle to stochasticity?

References:
All references are same as those cited in paper.

**Experience Assessment:**

I have read many papers in this area.

**Review Assessment: Checking Correctness Of Derivations And Theory:**

N/A

**Review Assessment: Checking Correctness Of Experiments:**

I carefully checked the experiments.

**Review Assessment: Thoroughness In Paper Reading:**

I read the paper thoroughly.

---

> ### Author Response · Authors · 2019-11-14
> **Author Response to Review #1 (part-1)**
>
> We thank the reviewer for providing a detailed feedback. As requested, we have conducted additional experiments to study the effect of performing multi-step predictions (i.e., K) (appendix B.6). Also, we have added additional baselines of 1-step predictions in all the main experiments (Figure 4-6 in the main paper). We wish to discuss some key points summarized as below.
>
> Q: "The novelty in their contribution is moderate - the idea of long-term future prediction is not new (e.g.: Ke et. al., 2019) (but using it for giving a curiosity bonus is new")
>
> >> Though there exists a rich literature studying the problem of long-term future prediction in RL, we wish to succinctly distinguish our work from the existing ones. First, most of the previous works build up long-term prediction model to derive a dynamics model for performing model-based planning/exploration, and thus deriving an accurate dynamics model becomes a major issue. However, in our work, our objective is to employ multi-step prediction to scale up the prediction difficulty from performing 1-step or self-prediction, which is critical for those prediction-error based curiosity methods. Second, while most of the existing long-term prediction models are formulated as autoregressive model with the base model performing 1-step forward prediction, the forward model in our work engages an open-loop multi-step prediction. The work from [Ke et. al., 2019] is a typical work that learns a long-term prediction model in an augoregressive manner for model-based planning, and thus be significantly different from ours.
>
>
> Q: "I expected a more detailed ablation for the choice of K, given that multi-step has major emphasis in the paper, but Figure 7(a) only shows ablations for 3 vs 1 step predictions." "- Only two values of K are tested - 1 and 3, what happens with larger values of K?"
>
> >> In our original submission, we thought that running the ablation study with multiple choices for K does not necessarily result in clear conclusion due to the variance in each run. Therefore, we only compared 3 vs 1 step in the paper, with the aim of giving a clear conclusion that inferring state novelty from 3-step forward prediction would have privilege over 1-step, as shown in Figure 7(a).
>
> To address the concern of the reviewer, we present an additional ablation study result that incorporates K values from the set of {1,2,3,4,5} in the revision (Appendix B.6). Generally, multi-step prediction with K=3 gives better performance than K=1/2. And we do not observe an apparent performance improvement when the value of K scales up to be greater than 3. Therefore, we suggest K=3 to be used over all the tasks, which seems to be a reasonable choice and could outperform K=1 in all cases.
>
>
> Q: "I feel that it may be the case that the 1-step version of the proposed model is actually good enough to beat all the baselines and adding multiple steps gives marginal gains. " "My major concern is that the gap between 1-step prediction and 3-step prediction (in Figure 7 (a)) is not significant. "
>
> >> The reviewer is right that the 1-step version of our model is actually good enough to beat all the baselines (in the testified domains). However, performing 3-step prediction gives noticeable benefit over the 1-step version of our model, which makes the modelling of multi-step prediction an essential part in our proposed method.
>
> Furthermore, to truthfully evaluate the novelty of our proposed work, besides comparing with our own variation of 1-step prediction, we suggest it's worth taking a look at the comparison result between our method and the typical 1-step forward prediction method ICM. Therefore, we present the actual convergence step (evaluated in number) for the ViZDoom tasks and show the result as below. The values shown in brackets correspond to the percentage of improvement for our 3-step model corresponding to the given baselines. Note that the amount of improvement from 3-step over 1-step is quite significant. Moreover, our 3-step model could outperform ICM baseline with significant margins. Therefore, with the given performance margins, we believe our proposed method would deserve to become a SOTA method over those challenging partially observable domains with sparse rewards.
> ================================================================
> Task                              ours(3-step)         ours(1-step)                ICM(1-step)
> ================================================================
> ViZDoom-dense           146 (-)		    188 (22.3%)		   350 (58.3%)
> ViZDoom-sparse	       	182 (-)		    255 (28.7%)		   601 (69.7%)
> ViZDoom-verySparse	197 (-)		    226 (12.7%)		   693 (71.6%)
> ================================================================

---

> ### Author Response · Authors · 2019-11-14
> **Author Response to Review #1 (part-2)**
>
> Q: (for RND) "The motivation for using long-term predictions to infer more meaningful novelty" is fine on it's own but seems to conflict with the choice of random network (RND) state features. Random features imply that a random "hash" of the observations is being computed which has no reason to have similar features for two nearby states."
>
> >> The reviewer raised an interesting point on using RND upon predicting forward transitions and we would like to discuss a bit more. RND in the original paper is developed to predict the input instead of forward transitions, with the intuition that predicting transition would result in a stochastic prediction function if there is *stochastic transitions like those involving randomly changing static noise on a TV* and thus is not desired. Therefore, it constructs a deterministic prediction function over the input.
>
> This intuition seems to be fine. However, we think the key point here is that even using RND over the input, it would not suffice to resolve the issues with extremely stochastic transitions like noisy-TV. The reason is that tasks like noisy-TV would impute stochastic noise at the input level, for which RND could not handle well with its "hash"-like effect. The experiment of RND is also conducted on tasks with moderately stochastic transition in Atari but not exactly on tasks with noisy-TV.
>
> In our paper, we tackle challenging partially observable domains with dense/rewards. The tasks we consider do not engage such stochastic transitions like noisy TV, and therefore, employing RND as a more efficient target function does not cause the model to be brittle. Furthermore, we demonstrate experiment result on Atari domains, which engage more stochastic transition than those navigation tasks. The results show that our model perform quite well. Also, we wish to rectify that for the partially observable domains, developing novelty model over predicting transitions is a more reasonable choice compared to predicting input itself, and employing RND has demonstrated to bring considerable benefit to policy training.
>
> We think developing robust exploration algorithms that can tackle stochastic transition cases like noisy-TV is absolutely a very important and promising research direction. We would like to explore more on it in our future work.
>
> We hope to know if our response has resolved the reviewer's concerns on RND. Please let us know if there is any further concerns.

---

### Official Review · AnonReviewer3 · 2019-10-25
**Official Blind Review #3**

**Rating:** 6

**Review:**

Sequence-level intrinsic exploration model for partial observable domains
This paper tackles the problem of RL in partially observable domains with sparse rewards. To address the sparse rewards issue, it proposes a sequence level intrinsic novelty model to guide policy learning. The sequence model is based on a dual-LSTM architecture. In general, this paper is well-written as easily accessible. Comprehensive experiments are provided to validate the effectiveness of the proposed methods.
The main issue with the paper is lacking discussions regarding the effective of the biased incur by the intrinsic reward. Specifically,
1)	How is the scaling factor beta determined? It would be nice if some discussions or experimental comparisons can be provided.
2)	The paper mainly deals with problems with sparse rewards. I wonder how the proposed method perform will in non-sparse rewards cases.  My main concern is that in the non-sparse reward cases, the intrinsic reward will cause bias, which may not guarantee good final performance.



**Experience Assessment:**

I have published in this field for several years.

**Review Assessment: Checking Correctness Of Derivations And Theory:**

N/A

**Review Assessment: Checking Correctness Of Experiments:**

I assessed the sensibility of the experiments.

**Review Assessment: Thoroughness In Paper Reading:**

I made a quick assessment of this paper.

---

> ### Author Response · Authors · 2019-11-14
> **Author Response to Review #3**
>
> We thank the reviewer for providing insightful comments. Bias incurred by incorporating intrinsic rewards is an interesting problem to investigate through and we would like to discuss on it in detail.
>
> Q: "How is the scaling factor beta determined?"
> >> We select the scaling factor from a range of {0.1, 0.01, 0.001, 0.0001, 0.00001} for each domain. Within such a range, we were able to find a value that works well for each of our evaluation domains.
> Ideally, if the scaling factor is too small, the reward bonus would be small so that its benefit on the policy training would be less apparent. Otherwise, if the scaling factor is too large, it would introduce bias in behavior and thus results in inferior task performance. We have conducted an additional ablation study to demonstrate the effect of scaling factor. We present a detailed analysis in Appendix B.5. We set the scaling factor from a range of {0.01, 0.001(ideal), 0.0001} in ViZDoom and investigate their effect on the "dense" and "very sparse" settings. The results of the ablation study provides a meaningful insight over the effect of the scaling factor on policy learning. The results genuinely agree with our previous assumption that setting the scaling factor to be too small/large would be less helpful in policy training.
>
>
> Q: "I wonder how the proposed method perform will in non-sparse rewards cases.  My main concern is that in the non-sparse reward cases, the intrinsic reward will cause bias, which may not guarantee good final performance"
>
> >> The results from ViZDoom-dense (Figure 4 in the main paper) as well as our ablation study in Appendix B.4. could reveal that introducing the intrinsic reward at a reasonable scale would not lead to severe bias to damage the task performance. In ViZDoom-dense, we employed the same scaling factor as the other two "sparse" domains, and the policy could still converge to 100% navigation success ratio. Furthermore, we add an additional result in a more complex policy training domain with dense rewards and large action space, Seaquest, from Atari 2600. We present a detailed analysis for this domain in Appendix E.2. Overall, our proposed method leads to significantly faster policy training progress than RND and ICM. This result shows that our proposed method could work well on dense-rewarded non-navigation tasks as well.
>
>
> We would like to know if our rebuttal adequately addressed the concerns of the reviewer. Please let us know if the reviewer would like to have additional discussions on the paper.

---

### Decision · Program_Chairs · 2019-12-19

**Decision:**

Reject

**Comment:**

This paper introduces a new architecture based on intrinsic rewards, to deal with partially observable and sparse reward domains.

The reviewers found the novelty of the work not particularly high, and had concerns about the general utility of the method based on the empirical evidence. This paper has numerous issues and could use significant revision in terms of writing, connections to literature, experiment design, and clarity of results.

Much of the discussion focused on the scaling parameter. From an algorithmic point of view, the scaling parameter is very problematic. It is domain specific and when tuned per domain resulted in very different values. The ablation study showed that only two settings in one domain led to good performance, whereas the other resulted in no learning (for some reason the other two values were not plotted).

There are concerns that the baselines were not completely fair. In many cases different domains were used to compare against RND and ICM, and there appears to be no tuning of these baselines for the new domains---this a problem due to the inherent bias in favor one's own method. In the solaris domain which was used in the RND paper, the results don't appear to match the RND paper, and in vizdoom the performance numbers are difficult to compare for ICM because a different metric is used---even if you don't like their performance numbers at least report them once so we can be confident the baselines are well calibrated. One reviewer pointed out the meta-parameters where different for RND than the published previous, but the paper does not describe what approach was used to tune those parameters and this is not acceptable. We cannot have much confidence that these results are reflective of those methods. Finally, there is no comment on how the performance numbers were computed and no description of how the errorbars where computed or what they represent.

The paper focuses on partially observable domains, the evidence that this method is effective in closer to Markov settings is unclear. The Atari experiments do not yield significant results by large (solairs looks as if there is no learning occurring at all---a no comment about it in the text to explain). The paper claims evidence the approach can work well in both cases, but it was not even indicated if frame-stacking was used in the Atari experiments. In fact, the result was only alluded too in the conclusion---there was no reference in the main text to a specific result in the appendix. Text is very challenging to read. The language is informal and imprecise, and the paper frequently uses terms incorrectly or in different ways through (e.g., the use of the term novelty throughout)

This is clearly an interesting direction. The authors should keep working, but this paper is not ready for publication. I urge the authors to dig deeper in the literature to gain a more nuanced understanding of the topic. Barto et al's excellent paper on the topic is a great place to start: https://www.ncbi.nlm.nih.gov/pmc/articles/PMC3858647/